# Laser Efficiency and Laser Safety: Holmium YAG vs. Thulium Fiber Laser

**DOI:** 10.3390/jcm12010149

**Published:** 2022-12-24

**Authors:** Alba Sierra, Mariela Corrales, Bhaskar Somani, Olivier Traxer

**Affiliations:** 1Urology Department, Hospital Clínic de Barcelona, Villarroel 170, 08036 Barcelona, Spain; 2GRC Urolithiasis No. 20, Tenon Hospital, Sorbonne University, F-75020 Paris, France; 3Department of Urology AP-HP, Tenon Hospital, Sorbonne University, F-75020 Paris, France; 4Urology Department, University Hospital Southampton NHS Trust, Southampton SO16 6YD, UK

**Keywords:** holmium YAG, thulium fiber laser, lithotripsy, laser settings, laser efficiency, laser safety, laser usability, kidney calculi, ureteroscopy

## Abstract

(1) Objective: To support the efficacy and safety of a range of thulium fiber laser (TFL) pre-set parameters for laser lithotripsy: the efficiency is compared against the Holmium:YAG (Ho:YAG) laser in the hands of juniors and experienced urologists using an in vitro ureteral model; the ureteral damage of both lasers is evaluated in an in vivo porcine model. (2) Materials and Methods: Ho:YAG laser technology and TFL technology, with a 200 µm core-diameter laser fibers in an in vitro saline ureteral model were used. Each participant performed 12 laser sessions. Each session included a 3-min lasering of stone phantoms (Begostone) with each laser technology in six different pre-settings retained from the Coloplast TFL Drive user interface pre-settings, for stone dusting: 0.5 J/10 Hz, 0.5 J/20 Hz, 0.7 J/10 Hz, 0.7 J/20 Hz, 1 J/12 Hz and 1 J/20 Hz. Both lasers were also used in three in vivo porcine models, lasering up to 20 W and 12 W in the renal pelvis and the ureter, respectively. Temperature was continuously recorded. After 3 weeks, a second look was done to verify the integrity of the ureters and kidney and an anatomopathological analysis was performed. (3) Results: Regarding laser lithotripsy efficiency, after 3 min of continuous lasering, the overall ablation rate (AR) percentage was 27% greater with the TFL technology (*p* < 0.0001). The energy per ablated mass [J/mg] was 24% lower when using the TFL (*p* < 0.0001). While junior urologists performed worse than seniors in all tests, they performed better when using the TFL than Ho:YAG technology (36% more AR and 36% fewer J/mg). In the in vivo porcine model, no urothelial damage was observed for both laser technologies, neither endoscopically during lasering, three weeks later, nor in the pathological test. (4) Conclusions: By using Coloplast TFL Drive GUI pre-set, TFL lithotripsy efficiency is higher than Ho:YAG laser, even in unexperienced hands. Concerning urothelial damage, both laser technologies with low power present no lesions.

## 1. Introduction

Ureteroscopy with laser lithotripsy is an extended surgical intervention used for urinary stone treatment [1]. The current gold standard laser is the Holmium YAG (Ho:YAG) laser. One of the latest technologies in laser lithotripsy is the thulium fiber laser (TFL) which uses a 10–20 µm silica fiber doped with elemental thulium to generate the laser beam. When compared to Ho:YAG technology, TFL technology results are more efficient, with an ablation rate of up to three times higher and a retropulsion value that is about three times lower [2,3,4]. Despite the consistent technological improvement in this field, there is a lack of consensus regarding the parameters to use [5] and a need for high-level evidence when it comes to TFL vs. Ho:YAG [6,7].

On the other hand, one of the major concerns with this new technology is its safety. Some authors believe that the more efficient absorption in water (1.9 μm for TFL and 2.1 μm for Ho:YAG) [8] may lead to more pronounced heating of the aqueous environment, causing indirect urothelial thermal injury [9,10,11]. To manage the double issue of safety management and choice of effective parameters, pre-settings might be helpful.

The aim of this study was to evaluate the relevance of low-power settings to manage effective stone dusting while maintaining safety during TFL lithotripsy. To test laser lithotripsy efficiency, a range of pre-settings as retained by Coloplast TFL Drive interface for stone dusting is compared between the holmium:YAG (Ho:YAG) laser and thulium fiber laser (TFL) in the hands of juniors and experienced urologists, using an in vitro ureteral model. Urothelial damage of both lasers was evaluated in an in vivo porcine model.

## 2. Materials and Methods

### 2.1. Laser Lithotripsy Efficiency

#### 2.1.1. Laser Systems

The Cyber: Ho 150 WTM (Ho:YAG laser) and the Fiber Dust (TFL) were used as laser generators. Both lasers were from Quanta System (Samarate, Lombardia, Italy). We chose those devices because the laser settings can be set identically in both laser generators (pulse energy and frequency). 

#### 2.1.2. Artificial Stones

We produced stone phantoms (5 mm cubes) according to previously described techniques [12]. Matching Begostone Plus powder (Bego France^®^, Villeurbanne, France) with distilled water, we aimed to reproduce calcium–oxalate monohydrate stones. A «powder to water» ratio of 15:3 was chosen, according to previous in vitro studies [13]. After confection, a drying period of 48 h at 30 °C was maintained to minimize the heterogeneity between stones. All stones were weighed with a digital balance (ASP-22E-001 Analytic Balance RADWAG serie) with 0.001 mg accuracy after the drying period.

#### 2.1.3. Experimental Setup 

The custom experimental setting, as previously described by [3], consisted of a ureteral model (polymer tube 17 cm length, closed on one side, 7 mm inner diameter), with an opaque tape on a tray with saline (Figure 1).

Trials were conducted using a single use digital flexible ureteroscope (Lithovue, Boston Scientific©, Maple Grove, MN, USA). Irrigation was ensured by a combination of a gravity irrigation at 40 cmH_2_O above the saline tray and a hand-assisted irrigation system providing on-demand forced irrigation to offer proper visibility, as is done in routine clinical practice (Figure 1). 

Participants were divided into two groups according to their skills (five junior urologists and five senior urologists performing more than 80 URS per year). Each one performed 12 continuous lasering sessions (6 with TFL and 6 with Ho:YAG laser) of 3 min with the following laser settings retained from the user interface pre-settings of the Coloplast TFL Drive for stone dusting: 0.5 J/10 Hz, 0.5 J/20 Hz, 0.7 J/10 Hz, 0.7 J/20 Hz, 1 J/12 Hz, 1 J/20 Hz. All tests were performed with a short pulse width from the manufacturer’s laser console settings and a 200 μm core-diameter silica fiber.

Data included laser settings (energy and frequency) and total energy. All stone fragments were labeled and dried at room temperature (21 °C). 

#### 2.1.4. Statistical Analysis 

SPSS v25 software (IBM Statistics, Chicago, IL, USA) was used for the statistical analysis. Ablation rates of different laser settings and equipment were recorded and analyzed. For each cohort of laser generators and set of laser parameters, 12 trials were performed. Results are presented as mean and percentages. To assess laser efficiency, one way ANOVA and T-student tests were used. A *p*-value of 0.05 or less was considered significant.

### 2.2. Urothelial Damage 

#### 2.2.1. Experimental Setup

Studies adhered to the Guide for the Care and Use of Laboratory Animals under an approval of Regional Animal Ethical Committee (#CEEA14). A French Government authorization and were conducted at CERC Faculté de Médecine Nord, Marseille, France (IACUC) (#D-13-055-22). 

Three female pigs were used for the experimentation (~40 kg). All procedures were performed under general anesthesia. The anesthetized pigs were placed in the dorsal position. A rigid cystoscopy was used to place a 0.035” guidewire (Terumo, Tokyo, Japan) into the pig kidney, and then a ureteral access sheath (UAS, Retrace 12/14 Fh, Coloplast, Denmark) was placed. A flexible ureteroscope was then positioned in the renal pelvis. An endoscopic evaluation of the renal pelvis was performed (Figure 2). We started the lasering in the renal pelvis, and then, in the distal ureter (after UAS removal), without touching the mucosa. 

#### 2.2.2. Laser Settings

The Cyber: Ho 150 WTM and the Fiber Dust generators (Quanta System Samarate, Lombardia, Italy) were used. Power limits were 20 W (1 J/20 Hz) and 12 W (1 J/12 Hz) in the kidney and ureter, respectively. All tests were performed with short pulse width and a 200 μm core-diameter silica fiber. In all pigs, TFL was used in the left ureter/kidney and Ho:YAG was used in the right ureter/kidney. We performed continuous lasering in the renal pelvis and in the renal papilla for 10 min in each area, to simulate the worst-case scenario. Then, we performed the same technique for 7 min (half on/half off) at the middle of the lumen of the proximal and distal ureter. Continuous irrigation was established at 40 cmH_2_O. 

#### 2.2.3. Method of Temperature Measurement

Before lasering, temperature was continuously recorded by a probe wire retrogradely inserted in the renal cavities that transmitted the intrarenal temperature into a console in the third pig.

#### 2.2.4. Post-Procedure Endoscopic Control

Urothelial injuries heal 5–10 days after their formation. To check tissue healing lesions, pigs were kept alive until a second ureteroscopy 18 days later. An endoscopic diagnosis exploration was made, and their kidneys and ureters were sent for analysis (Figure 2). Pathological analysis was performed by an independent laboratory to assess healing/fibrotic process.

## 3. Results

### 3.1. Laser Efficiency 

After 3 min of continue lasering, the overall ablation rate (AR) percentage of Begostone was 27% (*p* < 0.0001) greater with the TFL than with the Ho:YAG technology (Table 1). When comparing each setting, the overall mean AR was also superior for all groups using the TFL technology. Despite the laser source, differences were also found regarding AR (*p* < 0.001); the more delivered energy, the higher the AR. Similar results were found when comparing energy per stone weight (J/mg). The overall J/mg was 24% (*p* < 0.0001) lower when using the TFL than the Ho:YAG (Table 2) lasers. 

When comparing per expertise, junior urologists performed worse than seniors in all the tests for both lasers, AR percentage and energy per stone weight, but both juniors and seniors performed better when using TFL technology. However, when comparing their performance using TFL versus Ho:YAG lasers, juniors improved more than seniors (15% more AR percentage and 17% less J/mg). 

For ureteral stone treatment, the recommended laser power setting is less than 12 W [14]. Suggested settings for ureteral stones have 24% (*p* < 0.0001) better AR percentage and 21% (*p* = 0.004) lower energy per weight (J/mg) with TFL technology.

### 3.2. Urothelial Damage

Left and right urinary tracts were treated by TFL and Ho:YAG, respectively. A total of 23 tests were performed using three female pigs. The four defined sites were the renal papilla, renal pelvis and proximal and distal ureter. Due to time constraints, the left distal ureter of the third pig could not be tested.

First evaluation was performed endoscopically during laser activation. After 10 min of continuous lasering in the renal pelvis and the papilla, some small hyperemic lesions were seen in all kidneys with both Ho:YAG and TFL with no subjective differences between lasers using same power settings (Figure 3). No lesions were observed after seven sequential minutes (half on/half off) in the ureter for both lasers. Moreover, the maximal powers used (12 W and 20 W for ureter and kidney, respectively) were not accompanied with per-procedure safety issues. There was no bleeding, no perforation and no carbonization. 

During the third pig’s laser lithotripsy, the temperature was continuously recorded. In the left kidney (TFL), the temperature increased from 31.5 °C to 36.8 °C after 1.5 min of lasering. For 3 min, we progressively decreased irrigation until it stopped completely reaching a maximum of 40.2 °C. With continuous irrigation, the temperature remained around 36 °C during lasering and decreased below 35 °C when lasering stopped. For the right kidney (Ho:YAG), the temperature quickly increased from 33 °C to 41 °C when irrigation was slowed down until it stopped completely and remained between 33 °C and 34.6 °C during irrigation.

The second evaluation was performed endoscopically three weeks after laser lithotripsy. To access the renal pelvis, a UAS was inserted. Unspecific white marks were found in all kidneys, located at the upper papilla or peri-papilla and the renal pelvis (Figure 3). After UAS removal, ureteral evaluation was performed, and no lesions were found in neither the proximal nor distal ureter. Of note, in the third pig, where the temperature test was performed and the irrigation was voluntarily reduced, Bellini tubules were visible in both kidneys. 

No differences were detected after anatomopathological evaluation for both lasers and only slight inflammation was seen in some cases. Regarding the renal parenchyma, all animals had interstitial nephritis on both kidneys and showed no differences between Ho:YAG and TFL. 

## 4. Discussion

According to our results, TFL technology is superior to the Ho:YAG laser with better AR. This is not the first time that the TFL had performed better than the Ho:YAG laser [2,3,8]. Preclinical studies have shown promising results with a more efficient stone ablation rate and a faster ablation speed with TFL [8]. At the same energy and pulse frequency settings, TFL technology produces a significantly lower retropulsion rate than the current Ho:YAG technology [4]. This can be explained by several of the TFL’s characteristics. For instance, the fourfold higher wavelength absorption by water may result in greater absorption of laser energy during laser lithotripsy and also explain its high ablation efficiency over any type of stone [15,16,17]. Additionally, when focusing on peak power and pulse shape, the Ho:YAG’s peak power is extremely variable. On the contrary, TFL exhibits a nearly rectangular flat-top pulse shape with an almost constant low peak power (500 W) at different settings. At equivalent energy settings, the pulse generated by the TFL in SP mode is longer and has a lower peak power than the one of the Ho:YAG laser in the long pulse and Moses pulse modes [4,15]. These characteristics have been confirmed in several clinical studies after the approval of the US Food and Drug Administration and European CE mark in 2019 and 2020, respectively [6,18]. Even if clinical experiences are still low, we are starting to see high-quality trials with this new technology. Ulvik et al. [7] have recently published the first prospective randomized trial, showing that the TFL is superior to the Ho:YAG laser in terms of the stone-free rate, shorter operative time and fewer intraoperative complications. 

In addition, TFL seems to be more worthwhile for learners. When comparing results based on expertise, junior urologists performed worse than experienced urologist in all tests for both AR percentage and energy/stone weight. However, despite that, juniors performed better when using TFL technology, showing a reduced learning curve and lack of need to constantly adapt to a continuously changing stone position. This can be explained by the lower degree of retropulsion with TFL, which helps to improve the precision and vision during stone ablation [19]. Several lab studies have shown that TFL has lower retropulsion than Ho:YAG laser [4,18,20], leading to a more efficient lithotripsy [21]. Several clinical trials have also shown that TFL is a safe and effective modality for laser lithotripsy because of the lower retropulsion and minimal complication rate [22,23,24,25].

Although Ho:YAG has demonstrate an excellent safety profile, being considered as the more successful laser, TFL wavelength (1940 nm) is closer to water absorption peak, which results in four-fold higher abortion than Ho:YAG. This facilitates higher absorption of energy and increased ablation efficiency [8]. However, this higher rate of energy transfer to the stone and the surrounding fluid could potentially lead to indirect thermal damage [9,10,11]. Recently, Belle JD et al. [11] have demonstrated, in an in vitro silicone kidney-ureter model, that high-power lasers are associated with a risk of complications from thermal damage and therefore advocate using rather conservative laser settings for ureteroscopy laser lithotripsy. According to previously published papers, temperature rises proportionally to power [25], and power limits are settled at 20–30 W and 10–15 W for the kidney and ureter, respectively, to avoid cellular thermal damage [3,9,10]. Our study results are in line with that statement. We remark that low power settings are safe for the urothelium, as we confirmed in our second look of the porcine kidney. Moreover, the use of saline irrigation during the procedure has shown to be critical to avoid excessive temperature rises, and studies evaluating temperature rise with both Ho:YAG and TFL have demonstrated a good safety profile when continuous irrigation was applied during laser activation [10,26,27]. Similar findings are described during our trial, we were continuously lasering with continuous irrigation at 40 cmH_2_O and temperature remained at 36.8 °C and 34.6 °C for TFL and Ho:YAG, respectively. 

Regarding laser effectivity, we tried to simulate a real-life scenario, but the first limitation was an incomplete simulation of actual laser lithotripsy conditions in a urinary tract. Such conditions included ureteral peristalsis, respiratory movements and convection, which plays major roles during laser lithotripsy in the ureter. However, the aim of our study was to compare different settings and the obtained ablation rate. Stone phantoms, rather than human stones, were used. We required samples of approximately uniform mass, geometry, and composition, which could not be achieved practically with human stones. The third limitation involved the BegoStones immediately absorbing water through cracks and pores, which would have influenced the results of dehydrated phantoms in water. It should also be mentioned that the so-called dry phantoms in our study had not been desiccated. However, we have a control stone that was submerged into the saline tray without lithotripsy treatment, and we stored the stones in similar conditions. When its weight was the same as before the experiment, we assumed that the rest were dried too. In the porcine model, we were not lasering to stones, but we simulated the worst-case scenario through 10 min continuous lasering in the same place. 

## 5. Conclusions

In vitro, laser lithotripsy efficiency is higher with the TFL than with the Ho:YAG laser. Indeed, despite low power settings, AR was significantly higher, and less energy was needed to ablate 1 mg of stone with the TFL. Interestingly, it seemed that junior urologists had a faster learning curve with the TFL than with the Ho:YAG laser. Concerning laser safety, both laser technologies are equally safe. We can conclude than the Coloplast TFL Drive GUI pre-set values are effective and safe when working with 20 W in the kidney and 12 W in the ureter. 

## Figures and Tables

**Figure 1 jcm-12-00149-f001:**
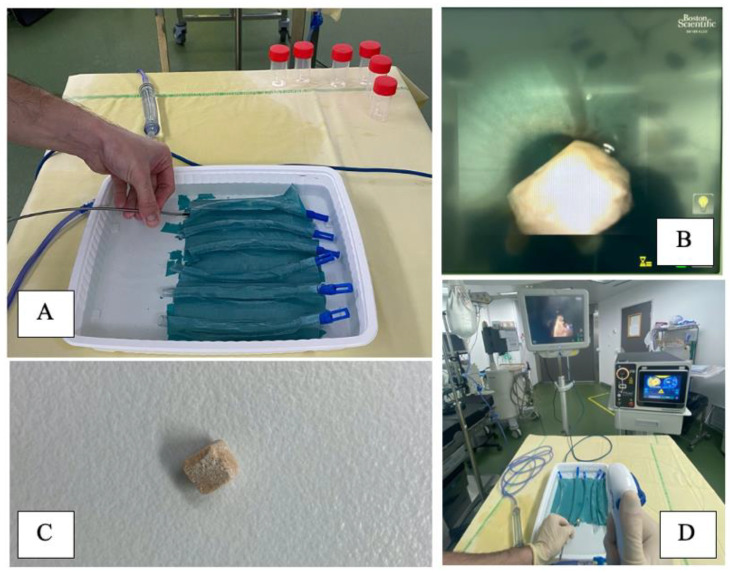
Experimental set up showing (**A**) Six polymer tubes, 17 cm length, closed on one side, 7 mm diameter, with an opaque tape on a tray with saline, used as a ureteral model (**B**) Endovision of the ureteral model with a Lithovue (Boston Scientific^®^, Maple Grove, MN, USA) and a BegoStone on it (**C**) BegoStone 5 × 5 × 5 mm^3^, dry weigh before and after the tests (**D**) Room display, using a Lithovue (Boston Scientific^®^, Maple Grove, MN, USA). Irrigation was ensured by a combination of a gravity irrigation at 40 cmH_2_O above the saline tray and a hand-assisted irrigation system.

**Figure 2 jcm-12-00149-f002:**
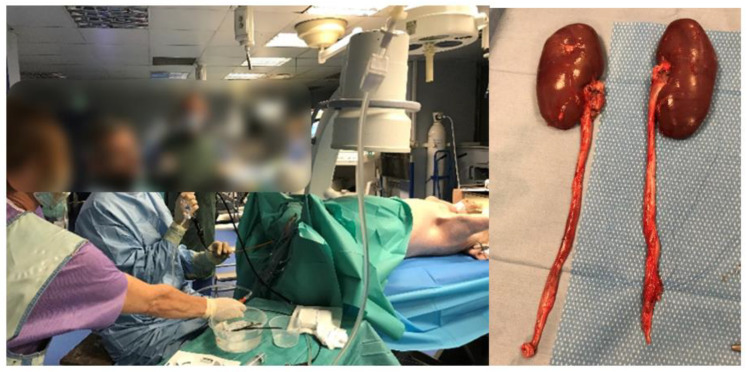
Female pigs (~40 kg). All procedures were performed under general anesthesia and the pigs were placed in the dorsal position. After the first procedure, pigs were kept alive and a second ureteroscopy was performed 18 days later to check for endoscopically tissue lesions. Animals were then sacrificed, and organs were removed for anatomopathological analysis.

**Figure 3 jcm-12-00149-f003:**
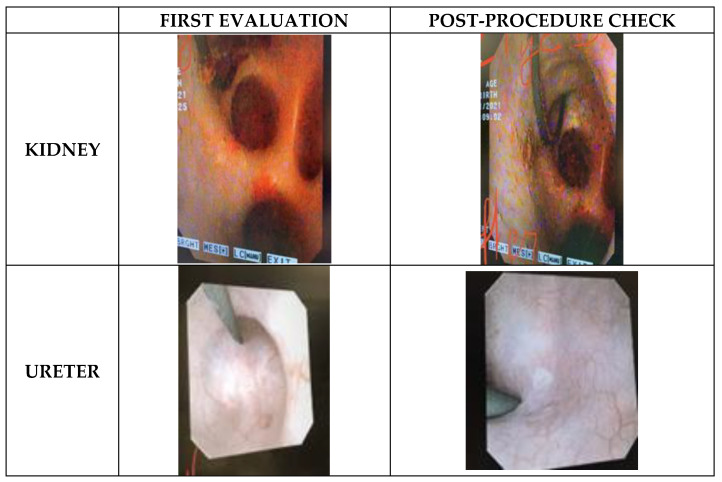
Endoscopic images captured during first evaluation and three weeks later, for kidney and ureter sites. Subjectively no differences were found endoscopically between TFL and Ho:YAG during per-procedure and post-procedure safety check.

**Table 1 jcm-12-00149-t001:** Ablation rate (mg/s) of each scenario by Ho:YAG and TFL lasers, during 3 min of laser lithotripsy. Holmium YAG (Ho:YAG). Thulium fiber laser (TFL).

			**Laser Settings**							
			**0.5 J/10 Hz ***	**0.5 J/20 Hz ***	**0.7 J/10 Hz ***	**0.7 J/20 Hz**	**1 J/12 Hz ***	**1 J/20 Hz**	**Ureter Tested Settings**	**All Tested Settings**
			Ablation rate (mg/s)							
**Junior** **(n = 5)**	**Mean**	**Ho:YAG**	8.24	10.04	17.12	28.18	41.26	41.44	19.165	24.38
		**TFL**	13.28	17.52	22.7	38.04	50.96	56.82	26.115	33.22
	**% Difference**		−52%	+61%	+74%	+32%	+35%	+23%	+37%	+36%
	***p* value**		0.10	0.1	0.03	<0.001	<0.0001	<0.001	<0.001	<0.0001
**Senior** **(n = 5)**	**Mean**	**Ho:YAG**	19.58	28.4	22.5	46.52	49.92	61.34	30.1	38.04
		**TFL**	26.08	31.2	30.44	59	58.56	70.82	36.57	46.02
	**% Difference**		−19%	+33%	+10%	+35%	+27%	+17%	+15%	+21%
	***p* value**		0.04	0.02	0.15	<0.0001	0.004	0.0001	<0.0001	<0.0001
**Total group** **(n = 10)**	**Mean**	**Ho:YAG**	13.91	19.22	19.81	37.35	45.59	51.39	24.63	31.21
		**TFL**	19.68	24.36	26.57	48.52	54.76	63.82	31.34	39.62
	**% Difference**		−33%	+41%	+27%	+34%	+30%	+20%	+24%	+27%
	***p* value**		0.002	0.008	<0.0001	<0.0001	<0.0001	<0.0001	<0.0001	<0.0001

* Coloplast TFL Drive user interface pre-settings for stone dusting in the ureter (≤12 W). Red front color means statistical significancy because is < 0.05.

**Table 2 jcm-12-00149-t002:** Comparison thulium fiber laser vs Holmium:YAG. Energy/stone weight (J/mg) in each scenario after 3 min of laser lithotripsy. Holmium YAG (Ho:YAG). Thulium fiber laser (TFL).

			**Laser Settings**							
			**0.5 J/10 Hz ***	**0.5 J/20 Hz ***	**0.7 J/10 Hz ***	**0.7 J/20 Hz**	**1 J/12 Hz ***	**1 J/20 Hz**	**Ureter Tested Settings**	**All Tested Settings**
			Energy/stone volume (J/mg)							
**Junior** **(n = 5)**	**Mean**	**Ho:YAG**	21.07	29.64	14.21	16.52	8.71	14.97	19.04	17.52
		**TFL**	10.03	19.25	10.26	10.11	7.25	10.48	13.34	11.23
	**% Difference**		−52%	−35%	−28%	−39%	−17%	−30%	−70%	−36%
	** *p* ** **value**		0.10	0.13	0.002	<0.0001	0.24	0.001	0.002	<0.0001
**Senior** **(n = 5)**	**Mean**	**Ho:YAG**	8.05	7.87	9.12	8.93	6.73	9.07	8.21	8.30
		**TFL**	6.52	10.63	7.82	6.68	7.08	7.14	8.21	7.64
	**% Difference**		−19%	+35%	−14%	−25%	+5%	−21%	0%	−19%
	** *p* ** **value**		0.04	0.02	0.05	0.06	0.77	0.04	0.99	0.07
**Total group** **(n = 10)**	**Mean**	**Ho:YAG**	14.77	19.04	12.61	13.29	8.06	12.31	13.62	13.35
		**TFL**	9.95	16.14	9.63	8.66	7.38	8.94	10.78	10.12
	**% Difference**		−33%	−15%	−24%	−35%	−8%	−27%	−21%	−24%
	** *p* ** **value**		0.06	0.17	0.0008	0.0002	0.32	0.001	0.004	<0.0001

* Coloplast TFL Drive user interface pre-settings for stone dusting in the ureter (≤12 W). Red front color means statistical significancy because is <0.05.

## Data Availability

Ask to the authors.

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
