# Peer review of "Laser Efficiency and Laser Safety: Holmium YAG vs. Thulium Fiber Laser"

_jcm, 2022, doi:10.3390/jcm12010149_

Round 1
Reviewer 1 Report
The manuscript covers in vitro and in vivo experiments on laser efficiency and laser safety of Thulium Fiber Lasers compared to Ho:YAG lasers.
The in vitro study is well thought out and performed. However, even more information and also results regarding ureteral damage can be found in a similar study of the authors published in the Journal of Endourology (Sierra A. et al.: Thermal injury and laser efficiency with Holmium:YAG and Thulium Fiber laser. An in vitro study’ DOI: 10.1089/end.2022.0216).
In my eyes, the goal of the in vivo study is very broad (‘to evaluate the safety of both lasers’). A clear hypothesis is not stated and therefore can’t be verified. One main point contributing to laser safety – the irrigation flow - is undetermined. This, however, is crucial with respect to a possible temperature increase. If the flow can’t be measured, it should at least be mentioned what the diameter of the working channel and inserted instruments were.
I think an exact in vitro measurement of the temperature increase under unambiguous conditions at different measured flow rates, as some research groups have already done, is much more meaningful. The author’s themselves did a good approach in this direction by assessing damages in an in vitro model (see publication mentioned above). Therefore, I wonder if animal experiments really do contribute additional information – especially, when the main pathological evaluation is performed after a three week healing period? I do not share the authors' opinion that this in vivo study, as cited in the abstract, demonstrates that 'laser safety' (?) is similar for both laser technologies. (I do not negate the statement itself, but the suitability of the study to prove it).
Unfortunately, since I don't think this article proves any new, relevant points, I don't support its publication.
Abstract
Line 16 ‘…retained from the user interface pre-settings of the Coloplast TFL Drive, for stone dusting: …’
Line 23 ‘The J/mg….’ à The energy per ablated mass [J/mg] was ….’
Line 25 ‘Laser safety was similar for both laser technologies in the in vivo porcine model.’
What exactly is meant with ‘laser safety’ (see general remark)
(Also: Line 27/28)
1. Introduction
Line 36 ‘long silica fiber’ ß please quantify
Line 36 ‘.. to generate the energy beam’ à ‘ … to generate the laser beam’
Line 38 To prove the higher removal efficiency of the TFL, only literature from the authors is cited. This point has also been investigated by work of other groups?
What is missing is reference to the work of other groups that have studied temperature elevation during lithotripsy as a function of irrigation flow (i.e., also size and occupancy of the working channel) (see e.g. Aldoukhi, A. H., Black, K. M., Hall, T. L., Ghani, K. R., Maxwell, A. D., MacConaghy, B., & Roberts, W. W. (2020). Defining thermally safe laser lithotripsy power and irrigation parameters: in vitro model. Journal of Endourology, 34(1), 76-81.)
2 Materials and Methods
2.1.1 Artifical stones
Line 62 What is meant by ‘0.001 accuracy’ – 0.1% accuracy or 0.001g= 1mg? Model and Manufacturer of the digital balance is missing.
Line 66 7mm = inner diameter?
Line 70 ‘… gravity irrigation at 40cm H2O’
I realize that the specification in this form is common. However, in my opinion, this leaves open how large the flushing flow is in the respective application situation (e.g. diameter and occupancy of the working channel). However, the flushing flow has a decisive influence on the temperature increase.
Line 78 ‘short pulse width’ ß please quantify (also in line 103)
2.2 Safety
See general remarks. This study is in my opinion not suitable to evaluate comprehensively ‘laser safety’ for all clinical situations and applications– e.g. it is unclear, what the irrigation flow or what the distance of the fiber to the tissue was. I think performing well thought through in vitro experiments showing under which conditions a critical temperature rise can occur will give better insight into what should and should not be done in the clinical application.
2.2.4 Post-procedure endoscopic control
It is unclear to me why the pathologic evaluation did not occur until after three weeks. Aren't superficial lesions / perforations already healed after this period?
3 Results
3.1 Laser efficiency
Line 137 ‘..the more delivered energy the more AR’
Do you mean: ‘ a higher power setting correlates with a higher ablation rate’?
3.2 Laser safety
Line 150 ‘due to the missing time calculation’ ??
After reading your article ‘Thermal injury and laser efficiency with holmium:YAG and Thulium fiber laser. An in vitro study’ I would like to remark: After cleaving a laser fiber, one must use a fiber stripper to remove the coloured plastic jacket. (I assume that with ‘transparent tip’ you refer to a correctly prepared fiber and with ‘cleaved LF’ you refer to a fiber that was cut within the coloured jacket without using a fiber stripper?)

Author Response
Dear reviewer, attached you can find the point-by-point response in a word document.
Thanks for you comments.

Reviewer 2 Report
The manuscript is very interesting and raises an important issue regarding the use of thulium laser fiber in the treatment of urolithiasis.
The study design is interesting with good methodology.
The advantage of the study is that both specialists and residents took part in it. The results achieved by residents on in vitro models allow for improvement and obtaining the best possible training models for trainees.
In addition, the added value is the use of fibers of the same diameter and the same settings of both lasers.
However, I would suggest addressing some issues:
· The paper lacks a paragraph about the study limitations, in which the authors should, among other things, raise issues related to the limitations of the in vitro model, such as the lack of respiratory movements, peristalsis, bends characteristic of the human ureter.
· "*" characters appear in the result tables, which are not explained anywhere.
· It is not precisely specified in the paper that in the case of assessing the safety of using lasers on pig models, lithotripsy was not performed, but only the laser was activated in the urinary tract. I suggest clarifying this information.
· In the paragraph: “Materials and Methods” subchapter 2.2. has too general a title - I suggest changing it to "laser safety" similar to subchapter 3.2 in the results.
· The sentence "It was not possible to check endoscopically the ureter" in line 166 is misleading. In the previous and soothing sentence, the authors describe the endoscopic evaluation of the urinary tract of pigs after 3 weeks and describe the changes present in the ureters, which is at odds with the quoted sentence. I suggest clarifying this statement.
Author Response

(The authors gave the same response as above.)

Reviewer 3 Report
Dear Authors,
First of all, I want to congratulate you on your brilliant idea. Ureteroscopy is a well-known procedure that is used mainly for ureteral lithiasis. Laser lithotripsy improved the stone-free rate. The thulium laser is expected to bring new possibilities for renal stones.
Here are my comments:
1. Please delete from Abstract - Objective, Materials and Methods, Results, and Conclusion.
2. l. 33-34. Please clearly define the role of ureteroscopy in treating renal and ureteral lithiasis and the benefits of laser lithotripsy.
3. l. 38-40, there is no subject in the sentence.
4. l. 73-74 performing 1.5 URS/week defines an experienced urologist?? It is probably a senior urologist - please clarify
5. l. 81 - the time of drying?
6. l. 98-99 lasering of what ???
7. l. 105-106 what was the distance of the laser to the renal papilla, the renal pelvis, and from the mucosa of the ureter? The results cannot be interpreted if the authors have not clearly defined the minimum distance.
8. l. 111-112 when the probe wire was inserted at the beginning of the procedure???
9. l. 117 what imaging technology was used?
10. l. 133-134, please define the ablation rate - AR. AR of what???
11. l. 140-143. the data presented are contradictory - junior performed worse than seniors....junior performed better - please clarify
12. l. 154 how the hyperemic lesions were subjectively evaluated???
13. l. 158 how do the authors evaluate the bleeding, perforation, and carbonization?
14. l. 160, the authors have not specified in the methods that the irrigation slowed down after 3 min. For how much time? How they slowed down the irrigation??
At what time after the laser stopped, the authors measured the temperature?
15. l. 159-164 - the temperature in the left kidney was initially 31.5, and in the right 33? Do the authors have an explanation for this?
16. l. 166 why was it impossible to check the ureter, the ureteroscope used has a 9-10 Ch diameter, and the ureteral access sheath was 12-14 Ch?
17. Figure 3. Images for the renal papilla and ureters are different. It is not the same level of ureter and papilla. Please provide different figures
18. l. 187 needs a reference
Author Response

(The authors gave the same response as above.)

Round 2
Reviewer 1 Report
Dear Editor,
I am afraid that I lack the medical knowledge to judge whether this work should be published. In my opinion, a very small cohort was examined here under conditions that are not precisely defined with a parameter set that is already used in clinical interventions on humans. Shouldn't the medical safety then already have been sufficiently proven during the approval procedure?
I therefore recommend that the work be submitted to yet another reviewer.
Reviewer 3 Report
Dear Authors,
Congratulations on your work. The manuscript has been improved, and I have no further suggestions.